

# Prognostic and therapeutic insights into colorectal carcinoma through immunogenic cell death gene profiling

Jinglu Yu[1], Yabin Gong[2], Zhenye Xu[3], Lei Chen[3], Shuang Li[4] and Yongkang Cui[4]

[1] PuDong Traditional Chinese Medicine Hospital, Shanghai University of Traditional Chinese Medicine, Shanghai, China, Shanghai, Pudong, China
[2] Department of Oncology, Yueyang Hospital of Integrated Traditional Chinese and Western Medicine, Shanghai University of Traditional Chinese Medicine, Shanghai, Xuhui District, China
[3] Department of Oncology, Longhua Hospital, Shanghai University of Traditional Chinese Medicine, Shanghai, Xuhui District, China
[4] Department of Gastroenterology, Baoshan Hospital of Integrated Traditional Chinese and Western Medicine, Shanghai University of Traditional Chinese Medicine, Shanghai, China

Corresponding authors
Shuang Li, lishuang6791@126.com
Yongkang Cui,
cuiyongkang2891@126.com

## ABSTRACT

While the significance of immunogenic cell death (ICD) in oncology is acknowledged, its specific impact on colorectal carcinoma remains underexplored. In this study, we delved into the role of ICD in colorectal carcinoma, a topic not yet comprehensively explored. A novel ICD quantification system was developed to forecast patient outcomes and the effectiveness of immunotherapy. Utilizing single-cell sequencing, we constructed an ICD score within the tumor immune microenvironment (TIME) and examined immunogenic cell death related genes (ICDRGs). Using data from TCGA and GEO, we discovered two separate molecular subcategories within 1,184 patients diagnosed with colon adenocarcinoma/rectum adenocarcinoma (COADREAD). The ICD score was established by principal component analysis (PCA), which classified patients into groups with low and high ICD scores. Further validation in three independent cohorts confirmed the model's accuracy in predicting immunotherapy success. Patients with higher ICD scores exhibited a "hot" immune phenotype and showed increased responsiveness to immunotherapy. Key genes in the model, such as *AKAP12*, *CALB2*, *CYR61*, and *MEIS2*, were found to enhance COADREAD cell proliferation, invasion, and *PD-L1* expression. These insights offered a new avenue for anti-tumor strategies by targeting ICD, marking advances in colorectal carcinoma treatment.

## INTRODUCTION

Colorectal carcinoma ranks as a leading gastrointestinal cancer and holds the third position in global cancer mortality. It presents a 5-year survival rate of near 65% (*Siegel et al., 2023*). Surgical resection remains the cornerstone for treating early-stage colorectal

cancer, while advanced stages often benefit from a regimen combining chemotherapy and targeted therapies (*Miller et al., 2022*). The advent of immune checkpoint inhibitors (ICI) has marked a revolutionary shift in oncological treatments, offering substantial clinical advantages (*Ciardiello et al., 2022*). Despite these advancements, the response to ICI is limited to a small patient subset (*Lenz et al., 2022*; *Gurjao et al., 2019*), underscoring the urgency to better predict outcomes and responses in colon adenocarcinoma/rectum adenocarcinoma (COADREAD), and to develop integrated approaches for overcoming immune resistance.

Historically, biomarker research in oncology predominantly utilized RNA sequencing (RNA-Seq) of whole tumor tissues (*Simoneau et al., 2021*), capturing only a generalized genetic snapshot from a diverse cell population. This approach resulted in ICI biomarkers with suboptimal predictive accuracy. The introduction of scRNA-Seq has transformed this area by allowing for the dissection of gene expression on a cellular level, opening up possibilities for discovering improved new biomarkers (*Wu et al., 2022*).

ICD is a form of programmed cell death that enhances adaptive immunity by considering its physical, chemical, and operational aspects (*Kroemer et al., 2022*). Tumor cell death, induced by external stimuli, releases various danger signals including high mobility group protein B1, surface-exposed calreticulin, type I interferon. These molecules convert non-immunogenic cells into immunogenic ones and serve as ligands for pattern recognition receptors. This interaction triggers the recruitment and activation of antigen-presenting cells, thereby initiating tumor-specific immune responses (*Hayashi et al., 2020*; *Haen et al., 2020*). Primary stressors inducing ICD include chemotherapy drugs, targeted agents, radiotherapy, and intracellular pathogens (*Salas-Benito et al., 2021*). Researches indicated that the synergistic application of chemotherapy and immunotherapy can enhance tumor cell immunogenicity, boost anti-cancer immune reactions (*Galluzzi et al., 2020*). This interplay positions ICD as a promising biomarker, intricately linked to patient prognosis and immunotherapy responsiveness. Previous studies have begun exploring ICD's clinical utility, particularly as a prognostic biomarker in cancers like clear cell renal cell carcinoma, melanoma (*Liu, Shi & Zhang, 2023*; *Ren et al., 2022*). However, there remains unknown in leveraging ICD for predicting responses to immunotherapy and chemotherapy in colorectal cancer. Addressing this gap was the focal point of our study, marking a step forward in this research domain.

In this investigation, our objective was to amalgamate single-cell and bulk RNA sequencing data to decipher the nuances of immune cell death in COADREAD patients, assessing its biomarker potential for predicting ICI outcomes. Additionally, we sought to elucidate the molecular and immune profiles associated with immune cell death and their implications for COADREAD prognosis. By analyzing immunogenic cell death related genes (ICDRGs), we have distinguished two subtypes and developed a robust predictive model, the ICD Score. This scoring system not only quantifies immune cell death but also holds promise as an instrumental marker in the realm of personalized precision medicine.

## MATERIALS AND METHODS

### Data sources and acquisition

The single cell data for this research was acquired from the GEO Database (https://www.ncbi.nlm.nih.gov/geo/) (*Barrett et al., 2013*), datasets GSE166555, which consisted of 12 samples of colorectal carcinoma. The TCGA database at https://portal.gdc.cancer.gov/ (*Blum, Wang & Zenklusen, 2018*) provided bulk RNA-seq data and clinical details for 380 COADREAD patients, GEO database also provided additional datasets, GSE17538 ($n = 238$) and GSE39582 ($n = 566$).

### Examining RNA sequencing at the individual cell level

The scRNA-seq data in R was analyzed using Seurat (v4.1.1) (*Mangiola, Doyle & Papenfuss, 2021*). Initial quality control was performed using the "Seurat" package to filter out cells that did not meet specific criteria. The criteria for selecting the dataset included nFeature_RNA between 200 and 7,500, nCount_RNA between 200 and 35,000, and a mitochondrial gene percentage below 10%. After these criteria, the dataset consisted of 66,050 cells and 21,753 genes. The dataset underwent normalization using the 'ScaleData' function to adjust for gene expression variability. Subsequently, PCA was performed on the top 2,000 highly variable genes to extract the top 28 principal components, serving as the basis for subsequent analysis. Post-PCA, the 'harmony' package was employed for batch correction on the PCA embeddings to mitigate batch effects, ensuring that the downstream analysis could accurately capture biological variations rather than technical discrepancies. Unsupervised clustering and visualization of cell subpopulations on a two-dimensional map was carried out using Uniform Manifold Approximation and Projection (UMAP) (*Becht et al., 2018*).

To compare gene expression differences between clusters, the 'FindAllMarkers' function was used. Marker genes for each cluster were determined based on a log2 (fold change) value greater than 1, an adjusted *p*-value lower than 0.05, and a resolution of 0.8. Finally, the cell subpopulations in the various clusters were annotated using the 'SingleR' package along with the CellMarker database (*Hu et al., 2023*) and the PanglaoDB database (*Franzén, Gan & Björkegren, 2019*).

Using the R function 'FindAllMarkers', we examined genes that were expressed differently in various cell types. In order to evaluate the diversity in sets of genes, we employed Gene Set Variation Analysis (GSVA) (version 1.40.1) (*Hänzelmann, Castelo & Guinney, 2013*). Scores were computed for 50 hallmark gene sets acquired from MSigDB (https://www.gsea-msigdb.org/gsea/msigdb) (*Liberzon et al., 2015*) and 34 ICD genes from the publication referenced by PMID 27057433 (*Garg, De Ruysscher & Agostinis, 2016*). Cell clusters were classified into high and low ICD score groups based on the median scores. Heatmaps were used to visualize the correlations between ICD and Hallmark scores, generated using the R packages 'pheatmap' and 'corrplot' with default thresholds.

### Bulk RNA-sequencing data integration

For the bulk RNA-seq datasets, we integrated the data and corrected batch effects using the 'limma' and 'sva' packages. Subsequently, a log2 transformation was applied to all datasets.

The integration process's effectiveness, before and after, was visualized using PCA plots, created with the 'FactoMineR' and 'factoextra' packages in R.

## Development of ICDRGs prognostic signature

We initially examined correlations between ICDRGs in COADREAD patients. Genes associated with prognosis were identified by univariate Cox regression analysis using the survival package. The analysis was conducted with a significance threshold of $p < 0.05$, as stated in the source (https://rdocumentation.org/packages/survival/versions/2.42-3).

## Molecular subtyping *via* consensus clustering

To categorize molecular subtypes based on ICD prognostic signature genes, we employed the 'ConsensusClusterPlus' package (1.59.0) in R. This involved utilizing the k-means algorithm for clustering with a range of k = 2–10 to determine the optimal cluster number. To ensure stability, this clustering process was iteratively conducted 1,000 times.

## ICDRGs prognostic signature development

By conducting univariate Cox regression analysis using the survival package at a significance level of $p < 0.05$, we identified genes linked to prognosis in COADREAD patients, while examining the correlations among ICDRGs.

## Molecular subtypes and clinicopathological correlation

We conducted an analysis to assess the clinical significance of molecular subtypes in COADREAD by examining their correlation with clinicopathological characteristics such as age, gender, grade, and TNM stage. The correlation was visualized through heatmaps using the 'pheatmap' package in R. Additionally, we performed survival analysis using the 'survminer' package to evaluate the prognostic implications of these subtypes.

## Pathway enrichment and immune infiltration analysis

In order to understand the TIME in different molecular subtypes, we employed GSVA using hallmark, KEGG, and Reactome gene sets obtained from the MSigDB database. By utilizing the 'estimate' package in R, the ESTIMATE algorithm (https://sourceforge.net/projects/estimateproject/) (*Scire et al., 2023*) computed stromal, immune, and ESTIMATE scores in tumor tissues. In addition, we utilized single-sample gene set enrichment analysis (ssGSEA) to investigate the immune infiltration pattern in COADREAD, specifically targeting 23 types of immune cells. The distribution of these immune cells in the samples was quantified and visualized using box plots.

## Molecular analysis and DEGs identification

The GSCA datasets (*Liu et al., 2023*) were used to analyze prognostic genes, including mutation, SNV, CNV, and Methylation. DEGs between clusters were identified using the 'limma' package, with a threshold of |log2 (Fold Change)| > 1 and an adjusted $p$-value < 0.05. Subsequently, the R package 'clusterProfiler' was employed for conducting GO and KEGG analyses. Significant DEGs with prognostic value were detected using univariate Cox regression and presented in a forest plot.

## ICD score system development

The ICD Score system was created through the use of PCA (*Ringnér, 2008*). This score represents the level of individual ICD regulation and is derived by summing the values of PC1 and PC2, which capture the most significant variances in the expression data. The ICD score was calculated for each sample, and an optimal threshold value was determined to maximize the accuracy of prognostic predictions. Based on this threshold, patients from the datasets were categorized into either low or high ICD score groups. Survival analyses were conducted to evaluate the prognostic significance of the ICD score in COADREAD. Box and bar diagrams were used to visualize the correlation between the ICD score and clinical features.

## Immune landscape analysis of ICD score

To demonstrate the connections between the ICD score and the infiltration of immune cells, as well as the expression of cytokines, chemokines, and their corresponding receptors, we utilized heatmaps. The relationship between the ICD score and hallmark pathways was evaluated using GSVA. Furthermore, we analyzed the mRNA expression of immune checkpoints and gene mutations in COADREAD patients from the TCGA database. We identified and visualized somatic mutations in different ICD score groups using waterfall plots and forest plots, providing insights into the genetic variations associated with ICD scores.

## Immunotherapy prediction and chemotherapy sensitivity analysis

The predictive value of the ICD score for immunotherapy response and chemotherapy sensitivity was assessed in our study. We analyzed the IMvigor210 cohort, which included 298 patients with urothelial carcinoma, to predict the immunotherapy response based on the ICD score. The data was processed using the R package 'IMvigor210CoreBiologies'. Additionally, we included the GSE61676 group, which consisted of transcriptomic data from advanced non-squamous non-small cell lung tumors treated with erlotinib and bevacizumab, to evaluate the predictive ability of the ICD score. Furthermore, information on the response of patients with kidney renal clear cell carcinoma (KIRC) to nivolumab (anti-PD-1) was obtained from Checkmate-KIRC (PMID 32472114) (*Braun et al., 2020*). Kaplan-Meier analysis was used to examine the prognostic significance of the ICD score in COADREAD. The ICD score was also employed to estimate the IC50 of commonly targeted therapeutic drugs using the 'pRRophetic' R package and data from the Genomics of Drug Sensitivity in Cancer (GDSC) database (*Yang et al., 2013*).

## RNA extraction and PCR analysis

Total RNA was extracted from colorectal cells using the Trizol method (Invitrogen), followed by cDNA synthesis using the Prime Script RT reagent Kit (Takara). Gene expression was quantified using TB Green Premix Ex Taq (Takara) and normalized to GAPDH. Table S1 provides primer details.

## Cell culture

Human colorectal cell lines including NCM460, HCT116, HT29, and LOVO were cultured in RPMI 1640 or DMEM (Gibco) supplemented with 10% FBS. The cells were incubated at 37 °C in a 5% $CO_2$ environment. To validate their authenticity, the cell lines obtained from the Chinese Academy of Sciences underwent STR profiling and mycoplasma testing, which can be verified at http://www.cellbank.org.cn/. The passage numbers of cell lines were strictly kept between 6 to 8. We routinely assessed morphology, growth rate, and adhesion, and conducted functional assays like drug sensitivity and migration to ensure cell line stability.

## siRNA transfection

siRNAs were used to target *AKAP12*, *CALB2*, *CYR61*, *MEIS2*, or a control, which were then transfected into HCT116 and HT29 cells. The cells were seeded at a density of $3 \times 10^5$ cells per dish (60 mm). Transfection was performed using Lipofectamine RNAiMax, followed by a 24-h culture in RPMI 1640 or DMEM supplemented with 10% FBS. The siRNA sequences can be found in Table S1.

## Proliferation assay

A total of 3,000 cells were placed in 96-well plates. The cells' viability was assessed every 24 h using a CCK-8 kit manufactured by Dojindo. The microplate reader (Thermo Fisher Scientific, Waltham, MA, USA) was used to measure the absorbance at 450 nm. The obtained data was then analyzed using GraphPad Prism 9.5.0.

## Transwell migration assay

A total of 100,000 cells were positioned in the top compartment on a membrane coated with Matrigel. RPMI 1640 or DMEM (without FBS) was added above, while a medium containing 20% FBS was placed below. Post 24-h incubation, cells were fixed, stained with Giemsa crystal violet, and imaged. Non-migrated cells from the upper chamber were removed.

## Statistical analysis

R software (version 4.1.2; *R Core Team, 2022*) was utilized for data analysis. Correlations were assessed using Pearson or Spearman methods, and the Wilcoxon test was employed to compare two groups. Survival outcomes among different subgroups were analyzed using Kaplan–Meier and log-rank tests. The prognostic significance of ICDRGs and clinicopathologic attributes was evaluated through univariate Cox regression. The Student's t-test was used to analyze the qRT-PCR results. A $p$-value less than 0.05 indicated statistical significance, denoted as *$p < 0.05$, **$p < 0.01$, and ***$p < 0.001$.

## Ethical considerations

The research did not involve human participants or animals as per the authors' confirmation.

## RESULTS

The flowchart of this study is shown in Fig. 1.

### Single cell sequencing data interpretation

#### Reducing complexity

Our analysis commenced with the single-cell sequencing dataset GSE166555, focusing on COADREAD. We integrated data from 12 distinct samples, ensuring minimal batch effect. Employing the Uniform Manifold Approximation and Projection (UMAP) algorithm, we segregated the cells into 28 identifiable clusters. By scrutinizing the expression of surface marker genes across these clusters, we pinpointed eight cell types: B cells, dendritic cells, mono-macrophages, natural killer (NK) cells, T cells, malignant cells, endothelial cells, and fibroblasts. These cell types were distinctively represented in different clusters, as illustrated in Fig. 2A.

### Transcriptomic landscape analysis

Our analysis revealed the most influential genes, with the top five contributors detailed in Fig. 2B. In malignant cells, the highest expressed marker genes were *KRT8*, *LGALS4*, *PIGR*, *ELF3*, and *CLDN4*, while the least expressed were *SRGN*, *VIM*, *TSC22D3*, *CXCR4*, and *RGS1*. For monocyte-macrophages, the most abundantly expressed markers included *AIF1*, *TYROBP*, *IL1-B*, *CXCL8*, and *S100A9*, contrasting with the lowest expressed markers *KRT8*, *LGALS4*, *PHGR1*, *PIGR*, and *JCHAIN*. In NK cells, the highest expressed genes were *TPSAB1*, *CPA3*, *MS4A2*, *HPGDS*, and *KIT*, with the least expressed being *KRT8*, *PHGR1*, *FXYD3*, *CLDN4*, and *IFI27*. T cells exhibited high expression of *CD3D*, *IL7R*, *CD2*, *CD3E*, and *TRAC*, and low expression of *LGALS4*, *KRT8*, *PIGR*, *IFI27*, and *CLDN3*. Gene Set Variation Analysis (GSVA) further delineated the Hallmark pathways enriched in each cell type, as shown in Fig. 2C. B cells and T cells demonstrated similar pathway enrichment patterns, while fibroblasts and monocyte-macrophages shared a different, yet comparable pattern. B cells and T cells demonstrated similar pathway enrichment patterns, while fibroblasts and monocyte-macrophages shared a different, yet comparable pattern.

### Expression profiles of ICDRGs

In our analysis, the set of 34 ICD-associated genes were referenced from literature (PMID: 27057433), enabling us to discern the expression patterns of ICDRGs across various cell types. These expression patterns were effectively visualized through bubble plots (Fig. 2D). Notably, *HSP90AA1* exhibited high expression in B cells, T cells, and NK cells, while showing lower expression in dendritic cells, monocyte-macrophages, malignant cells, endothelial cells, and fibroblasts. Conversely, *HMGB1* was predominantly expressed in NK cells, T cells, and fibroblasts, but less so in B cells, dendritic cells, monocyte-macrophages, malignant cells, and endothelial cells. *CALR* expression was significantly elevated in B cells compared to other cell types. *PDIA3* showed a similar pattern, with heightened expression in B cells and NK cells, but lower levels in other cell types. Additionally, *IFNGR1*, *CD4*, *LY96*, and *IL1B* were predominantly expressed in monocyte-macrophages, while *IFNGR1* and *CD4*, along with *MYD88*, were highly expressed in dendritic cells.
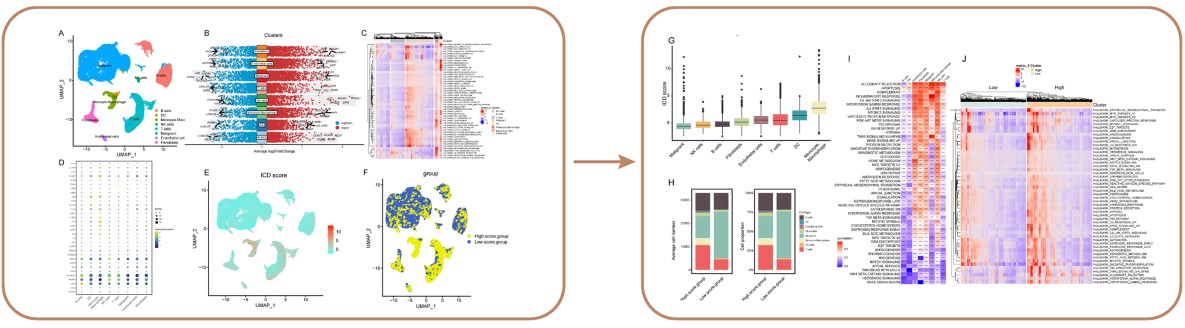

Construction of the ICD score at single cell level

Characteristics of the ICD score at single cell level

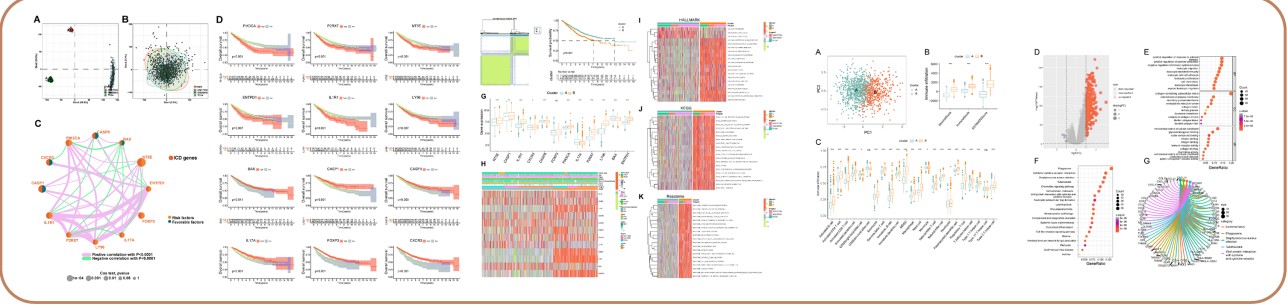

Construction and landscape of the ICD molecular clusters

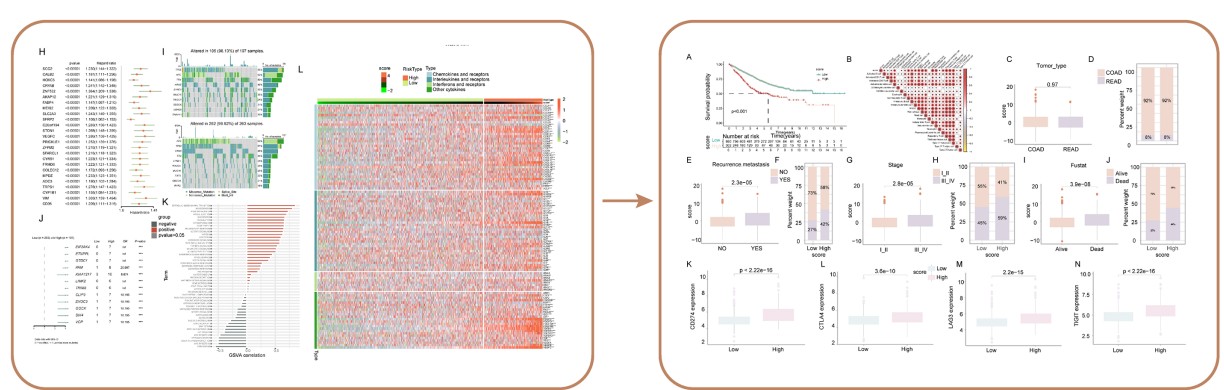

Construction of the ICD score

landscape of the ICD score

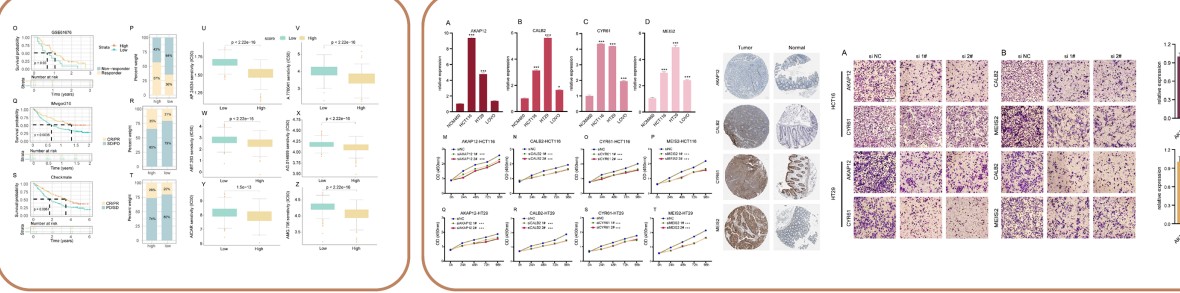

Immunotherapy and drug susceptibility

Experiment verification

**Figure 1 Flowchart of this study.**

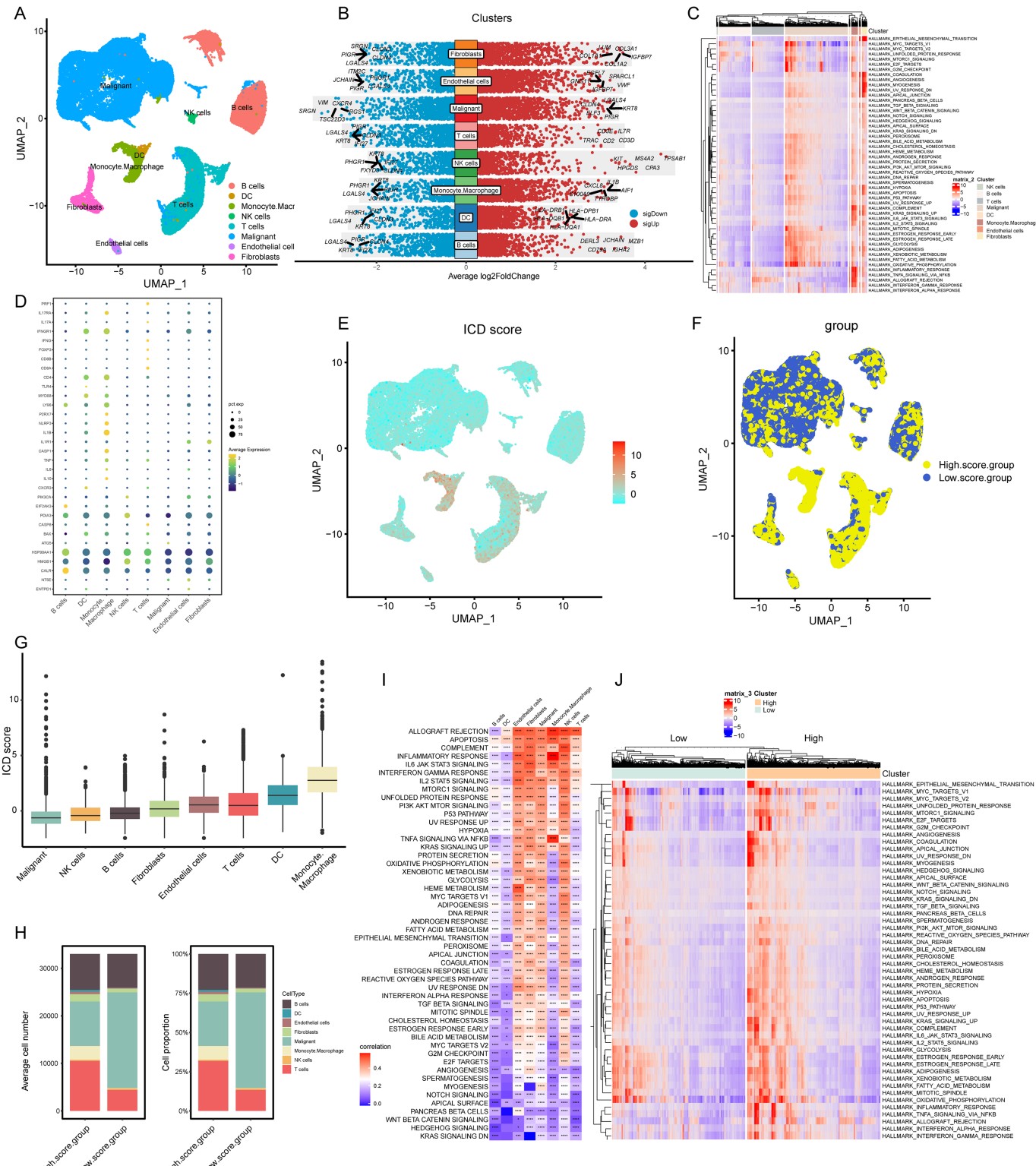

**Figure 2 Elaboration of ICD score and cellular diversity in COADREAD *via* scRNA-seq.** (A) The COADREAD cell dataset was delineated into eight distinct clusters *via* UMAP, featuring varied cell types: B cells, dendritic cells, endothelial cells, fibroblasts, malignant cells, mono-macrophages, NK cells, and T cells. (B) A volcano plot delineated differentially expressed genes among these cell types, highlighting the top five. (C) Displayed through a heatmap, this illustrated the enrichment levels of Hallmark pathways in each tumor-infiltrating cell type, with a bar indicating the relative

**Figure 2 (continued)**
pathway enrichment. (D) A dot plot revealed the expression levels of 34 ICDRGs across these varied cell types. (E) GSVA was used to compute the ICD score for each cell type, visualized through a UMAP plot indicating the expression of ICD gene clusters. (F) Cell clusters were categorized into high and low ICD score groups, illustrated *via* UMAP plot. (G) Box plots represented the ICD scores across cell clusters, organized in ascending order. (H) A composition chart compared the number and proportion of cells in high *versus* low ICD score groups. (I) A heatmap demonstrated the relationship between ICD score and Hallmark pathway scores across all clusters. (J) This heatmap contrasted Hallmark pathway scores between high and low ICD score groups.                               

## ICD score elucidation and implications

In our analysis, cells were scored for expression of ICDRGs using the GSVA method, as shown in Fig. 2E. This led to categorization into low and high ICD score clusters (Fig. 2F), with the score distribution visualized in box plots (Fig. 2G). Monocyte-macrophages exhibited the highest ICD scores, in contrast to tumor cells, which scored the lowest. Composition charts displayed the cell type distribution in these clusters, based on count and proportion (Fig. 2H), highlighting a prevalence of tumor cells in low-score and T cells in high-score clusters.

Correlations between ICD scores and Hallmark pathway scores (Fig. 2I) revealed positive associations with allograft rejection, complement, and inflammatory response pathways in monocyte-macrophages, NK cells, and endothelial cells. B cells, dendritic cells, and T cells showed negative correlations with pathways like KRAS signaling DN and WNT beta-catenin signaling. High TLS score group exhibited elevated expression of immune activation pathways (Fig. 2J), with inflammatory response associated pathways markedly enriched.

## Comprehensive analysis of bulk-seq data

Data from 1,184 patients were aggregated from the TCGA (TCGA-COADREAD) and GEO databases (GSE39582, GSE17538). We addressed batch effects using the "limma" and "sva" methods. Figures 3A and 3B showed a notable reduction in the dimensions of components one and two from 76.3% to 15.6%.

## Delineation of ICD molecular subtypes and their clinical relevance

Our survival analysis identified 12 ICDRGs-*PIK3CA, P2RX7, NT5E, ENTPD1, IL1R1, LY96, BAX, CASP1, CASP8, IL17A, FOXP3, CXCR3*-significantly correlated with overall survival (OS) in COADREAD patients (Fig. 3D, $p < 0.05$). Univariate Cox regression highlighted that *PIK3CA, P2RX7, NT5E, ENTPD1, IL1R1, LY96* positively impacted survival, whereas *BAX, CASP1, CASP8, IL17A, FOXP3, CXCR3* had a negative association. We constructed an ICD network to elucidate these genes' interrelationships and prognostic significance (Fig. 3C).

COADREAD patients were divided into two molecular subtypes, A and B, based on ICDRG expression profiles using a consensus clustering algorithm. Optimal separation (k = 2) indicated distinct survival probabilities between these subtypes, with subtype A showing higher survival (log-rank test, $p = 0.001$; Fig. 3F). Most ICDRGs were highly expressed in B cluster (Fig. 3G). Additionally, the variances between ICDRG expression and clinical features like age, stage, gender, recurrence, metastasis, fustat, and futime across

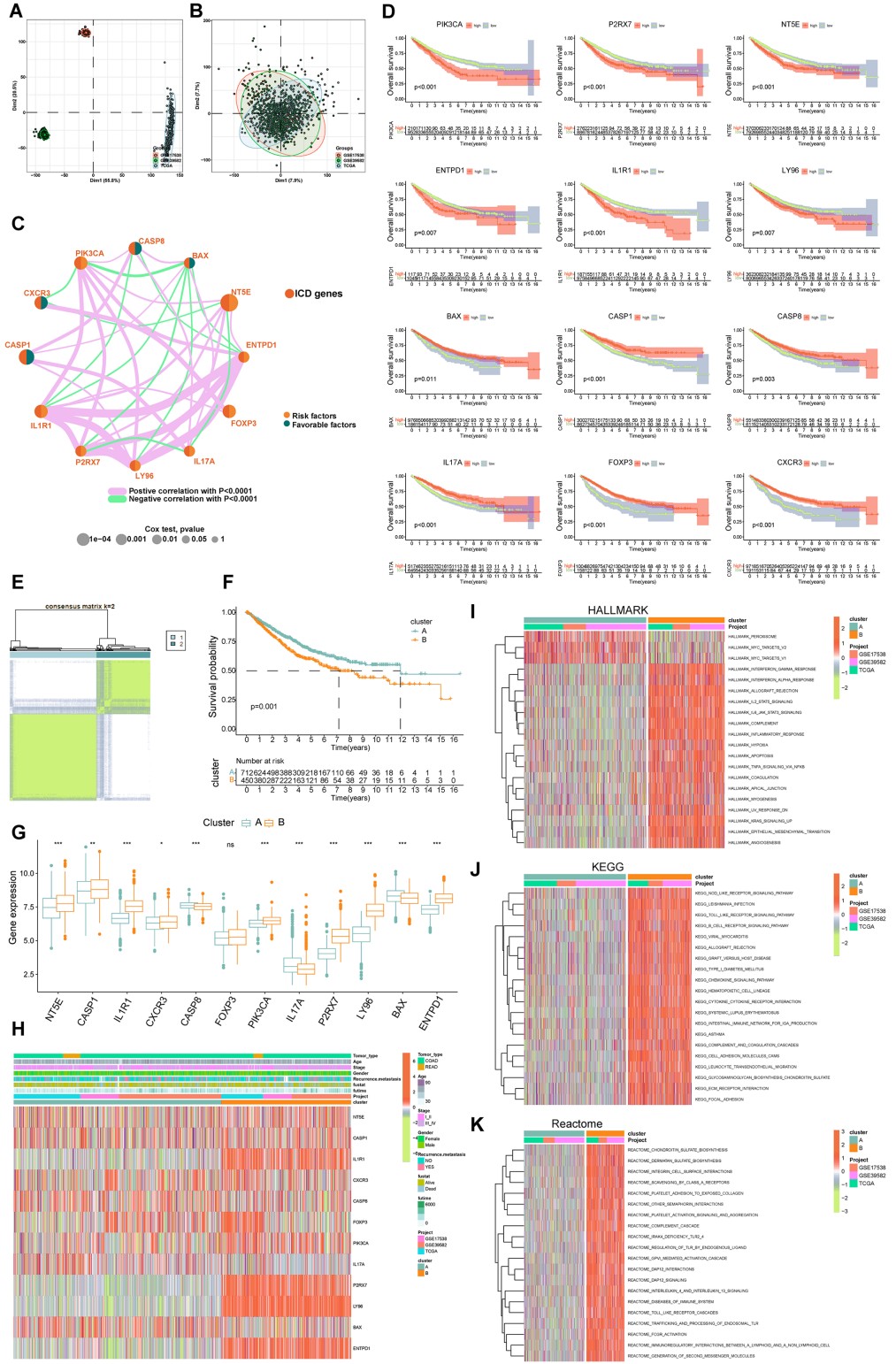

**Figure 3 Delineation and biological characterization of ICD molecular subtypes in colorectal carcinoma.** (A) Illustrating the distribution of transcriptional expressions of 34 ICDRGs in three colorectal adenocarcinoma cohorts. (B) Enhanced visualization post-adjustment, clarifying transcriptional diversity. (C) Spearman correlation analyses depicted the interactions among ICDRGs in 1,184 colorectal

**Figure 3 (continued)**
cancer samples, with line thickness indicating correlation strength-pink for positive and green for negative. (D) Log-rank tests evaluated the survival impact of specific ICDRGs (PIK3CA, P2RX7, NT5E, ENTPD1, IL1R1, LY96, BAX, CASP1, CASP8, IL17A, FOXP3, CXCR3). (E) Consensus clustering analysis revealed two distinct subtypes (k = 2) in colorectal cancer samples. (F) Highlights significant survival differences between molecular subtypes A and B. (G) Analyzed the abundance of ICD signature genes within the two identified clusters. (H) A heat-map demonstrated the association of clinicopathologic characteristics with each subtype, using red to indicate up-regulation and green for down-regulation. (I–K) GSVA analysis of Hallmark (I), KEGG (J), and Reactome (K) pathways between molecular subtypes A and B, with red denoting more enriched and green denoting less enriched pathways. $p$-value denoted as $*p < 0.05$, $**p < 0.01$, and $***p < 0.001$.

the two molecular subtypes had no significant differences (Fig. 3H). Chi-square test and Wilcoxon test were used for the comparison.

## Analyzing ICD subtypes through GSVA

GSVA enrichment analyses on molecular subtypes using Hallmark, KEGG, and Reactome pathways showed subtype B's significant enrichment in inflammation-related pathways, including interferon gamma and alpha, IL2-STAT5, IL-6-JAK-STAT3, and Toll-like receptor and B cell receptor signaling (Figs. 3I–3K), further establishing the link between subtype B and inflammation, a key aspect of immunogenic cell death.

## ICDRGs analysis *via* GSCA database

We analyzed ICDRGs through the GSCA database (http://bioinfo.life.hust.edu.cn/GSCA/#/). The analysis covered SNV percentages (Fig. S1A), mutation frequencies (Fig. S1B), CNVs (Fig. S1C), and correlations between CNV, methylation, and mRNA expression (Figs. S1D–S1F) in COADREAD. This provided insights into genetic and epigenetic variations of ICD signature genes.

## Investigating immune infiltration in ICD subtypes

PCA analysis demonstrated distinct ICD transcription profiles between subtypes A and B (Fig. 4A). Investigating ICDRGs' roles in the TIME of COADREAD, we found subtype B showed higher stromal, immune, and ESTIMATE scores (Fig. 4B). The estimate algorithm was used to calculate human immune cell subsets for each COADREAD sample in immune cell infiltration between the subtypes *via* ssGSEA (Fig. 4C). Specifically, cells like activated B and CD4 T cells, dendritic cells, MDSC, macrophages, and various T-helper cells showed lower infiltration levels in subtype A compared to subtype B. These findings suggested that subtype B is enriched in immune pathways and closely associated with TIME.

## Exploring ICD-DEGs: functional and pathway enrichment

We identified 448 differentially expressed genes (DEGs) between ICD molecular subtypes A and B, visualized in a volcano plot (Fig. 4D, |logFC| > 1, $p < 0.05$). GO and KEGG analyses were conducted to identify associated biological pathways. GO enrichment revealed DEGs predominantly involved in cytokine production regulation, leukocyte

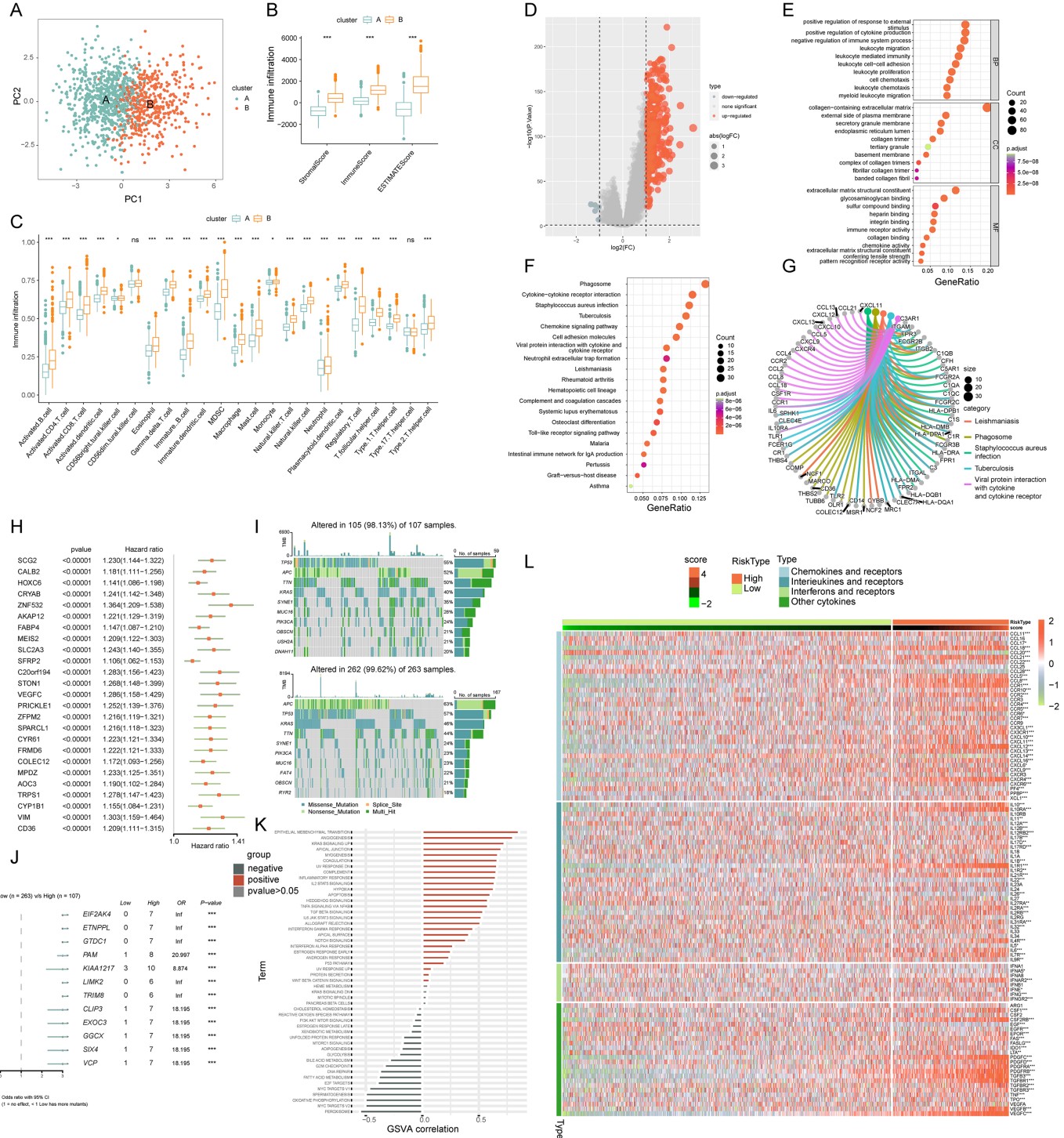

**Figure 4  Comprehensive ICD score analysis in colorectal carcinoma.** (A) PCA illustrating notable distribution differences among ICD molecular subtypes. (B) Comparison of stromal, immune, and ESTIMATE scores in different ICD clusters. (C) Analysis of immune cell infiltration in clusters A and B. (D) Volcano plot presenting 448 ICD-DEGs from varied molecular subtypes. (E) GO enrichment analysis for ICD-DEGs, with plot size indicating gene counts, categorized by BP, CC, and MF. (F) KEGG enrichment analysis of ICD-DEGs, detailing gene enrichment. (G) Chord graph highlighting five key KEGG pathways and associated genes. (H) PCA-based ICD score construction from 25 ICD-DEGs and subsequent survival analysis of two score clusters. (I) Waterfall plots of somatic mutations in high and low ICD score groups. (J) Forest plot comparing mutational variation and HRs in high and low-score clusters. (K) GSVA correlation of ICD score with 50 hallmark pathways. (L) GSVA for cytokines, chemokines, and their receptors in high and low ICD score clusters. *p*-value denoted as **$p < 0.01$ and ***$p < 0.001$.

migration and mediated immunity, cell and leukocyte chemotaxis, immune receptor and chemokine activity, and pattern recognition receptor pathways (Fig. 4E). KEGG enrichment highlighted DEGs in cytokine-cytokine receptor interaction, chemokine signaling, viral protein interaction with cytokine and cytokine receptor (Fig. 4F). A cnetplot elucidated specific gene networks in these pathways, with the top five pathways shown in Fig. 4G. This enrichment analysis suggested ICD's significant role in immune activation and cytokine-chemokine interactions in COADREAD.

### Development and implications of ICD score

Univariate Cox regression analysis was employed on the 448 DEGs to assess their prognostic significance, leading to the identification of 25 OS-related genes. These genes were represented in a forest plot (Fig. 4H). Subsequently, PCA was conducted on these 25 prognostic ICD-DEGs, dividing patients into low and high ICD score genomic subtypes (score = PCA1–PCA2).

### Examining the association with somatic mutations

The somatic mutations in the two ICD score clusters were further analyzed. The top 10 mutated genes in the high ICD score cluster included *TP53*, *APC*, *TTN*, and *KRAS* (Fig. 4I upper), while the low ICD score cluster showed a different gene mutation profile (Fig. 4I under), indicating a higher overall mutation frequency in the high score cluster. A forest plot highlighted the top 12 genes with the most significant variation in mutation frequency between the clusters (Fig. 4J). This analysis suggested a positive correlation between higher gene mutation frequency and variation in the high ICD score cluster.

### ICD score correlation with TIME and pathways

The relationship between the ICD score and hallmark pathway scores, analyzed using GSVA, revealed a positive correlation with immune activation pathways, including interferon gamma and alpha responses, inflammatory response, IL-2-STAT5 signaling, and TNFA signaling *via* NFKB. Conversely, a negative correlation was observed with pathways related to malignancy like peroxisome, MYC targets, oxidative phosphorylation, and the reactive oxygen species pathway (Fig. 4K). This suggests the ICD score's alignment with immune-activated pathways.

Further analysis of cytokine-chemokine networks highlighted significant enrichment of chemokines, interleukins, and interferons along with their receptors in the high-ICD cluster (Fig. 4L). Spearman correlation analysis indicated a positive correlation of the ICD score with the infiltration of 23 immune cell types, suggesting a more robust immune component in the TIME of the high-score cluster and potentially better immune prognosis (Fig. 5B).

### ICD score's impact on COADREAD prognosis

Survival analysis revealed that COADREAD patients in the low-score ICD cluster exhibited higher survival rates than those in the high-score cluster (Fig. 5A, $p < 0.001$). Significant variations in recurrence, metastasis, and disease stage were observed between the clusters, with advanced stages showing higher ICD scores (Figs. 5C–5J). Further

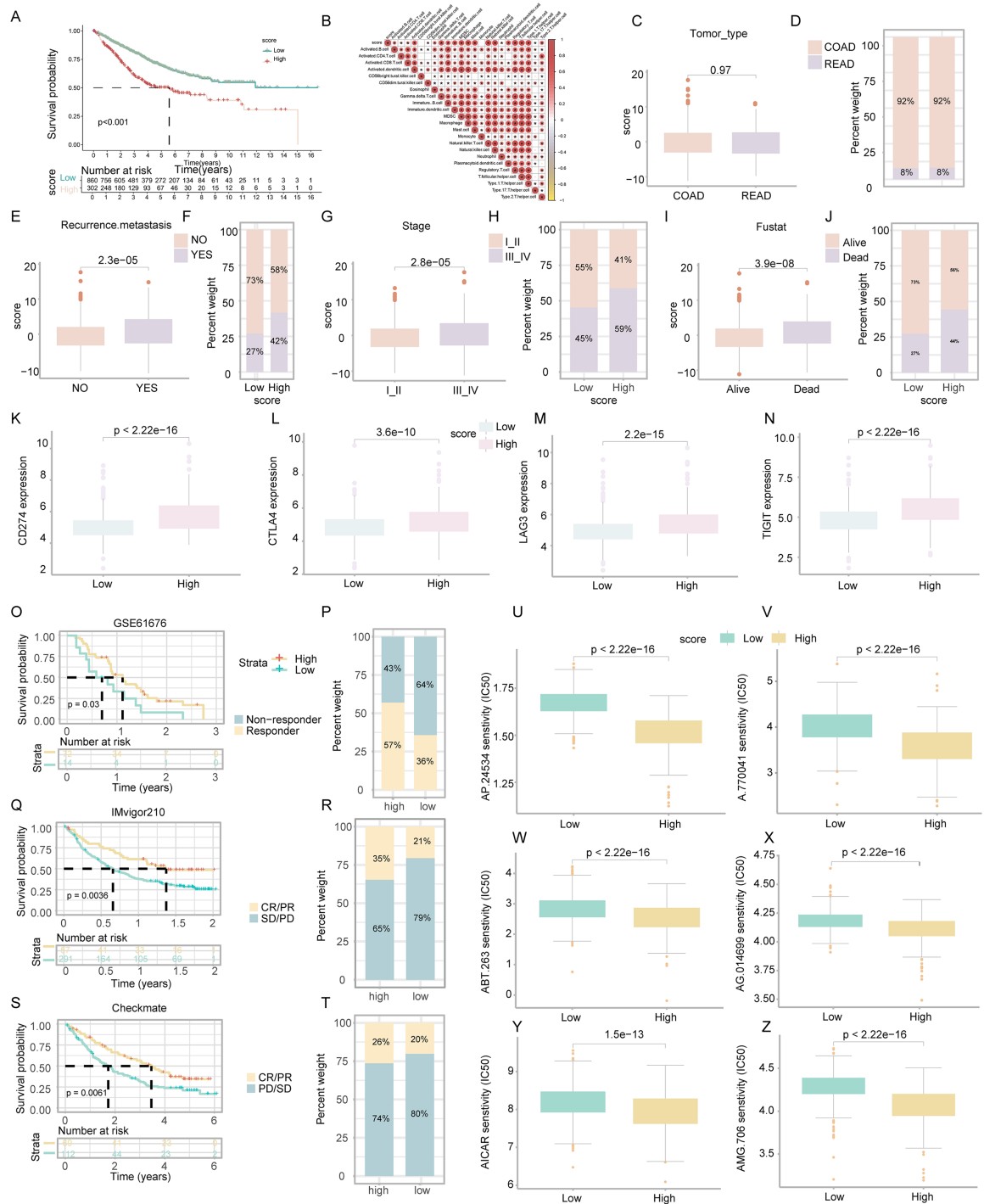

**Figure 5 Assessment of clinical impact, immunotherapy response, and drug sensitivity in relation to ICD score.** (A) Survival analysis examined the prognostic differences between high and low ICD score groups. (B) Analysis of the positive relationship between ICD score and tumor-infiltrating immune cells. (C–J) Investigates the differences in tumor type (C, D), recurrence and metastasis (E, F), stage (G, H), and survival status (I, J) across ICD score clusters. (K–N) Examined the variance in immune checkpoint levels between high and low score groups. (O, P) Targeted therapy response in lung cancer (GSE61676). Comparison of responses and survival outcomes between low and high ICD score clusters. (Q, R) Immunotherapy efficacy in urothelial carcinoma (IMvigor210). Analysis of response and survival differences in low *versus* high ICD score clusters. (S, T) Anti-PD-1 therapy response in KIRC cohort (Checkmate). Evaluated therapeutic response and survival rates between low and high ICD score groups. (U–Z) Box diagrams depicting the differences in drug sensitivity to targeted therapy between high and low ICD score clusters.

analysis demonstrated elevated expression of immune checkpoints like *PDCD1*, *CTLA4*, *TIGIT*, and *LAG3* in the high-score group (Figs. 5K–5N), suggesting a potential for better response to immune checkpoint inhibitors in this group. This underlined the ICD score's relevance in predicting immunotherapy efficacy and regulating the tumor immune microenvironment in COADREAD.

### Efficacy of ICD score in predicting response to immunotherapy

We assessed the prognostic value of the ICD score for immunotherapies, including PD-L1 blockade, in various cohorts. In the GSE61676 cohort, a higher response rate to bevacizumab combined with erlotinib was observed in the high-ICD cluster (57%) compared to the low cluster (36%) (Fig. 5P). This cluster also showed significantly longer overall survival (OS) (Fig. 5O, $p = 0.03$). In the IMvigor210 cohort, patients' responses to anti-PD-L1 therapy varied. Patients with stable or progressive disease had lower ICD scores compared to those with complete or partial responses (Fig. 5R). The high-ICD cluster demonstrated significant clinical benefits and longer OS (Fig. 5Q, $p = 0.0036$). Similarly, in the Checkmate cohort, the high-ICD cluster had significantly longer OS (Fig. 5S, $p = 0.0061$) and a higher percentage of complete/partial responses (Fig. 5T). These findings indicated that the ICD score was a potential predictor of immunotherapy efficacy in COADREAD.

### Drug sensitivity analysis based on ICD score

Patients in the high-score cluster exhibited lower IC50 values for drugs like AP.24534, A.770041, ABT.263, AG.014699, AICAR, and AMG.706 (Figs. 5U–5Z), indicating a higher sensitivity to these treatments. These findings suggested the utility of the ICD score as a predictor for selecting effective anticancer drugs.

### ICDRGs' role in COADREAD cellular behaviors

Our research involved RT-qPCR analysis of *AKAP12*, *CALB2*, *CYR61*, and *MEIS2* in normal colon and COADREAD cell lines, revealing significant overexpression in tumor cells (Figs. 6A–6D). We further investigated these genes' roles in COADREAD using siRNA knockdown in HCT116 and HT29 cells. The siRNA sequences for target genes *AKAP12*, *CALB2*, *CYR61*, and *MEIS2* are listed in Supplementary_Material 1, with siRNA-1 and siRNA-2 selected for their high transfection efficiency exceeding 70% compared to negative controls (Figs. S2A–S2H). The impact on cell proliferation and invasion was assessed through CCK8 and Transwell assays, respectively. Knockdown of these four genes markedly inhibited proliferation and invasion in both cell lines (Figs. 6E–6N). Additionally, their knockdown significantly reduced PD-L1 expression, suggesting potential in combining ICD inducers with PD-L1 inhibitors (Figs. 6P and 6Q). Immunohistochemical analysis from the HPA database further confirmed higher protein expression levels of these genes in COADREAD stroma (Fig. 6O).

In summary, *AKAP12*, *CALB2*, *CYR61*, and *MEIS2* were key regulators significantly influencing the biological behaviors of COADREAD cells.

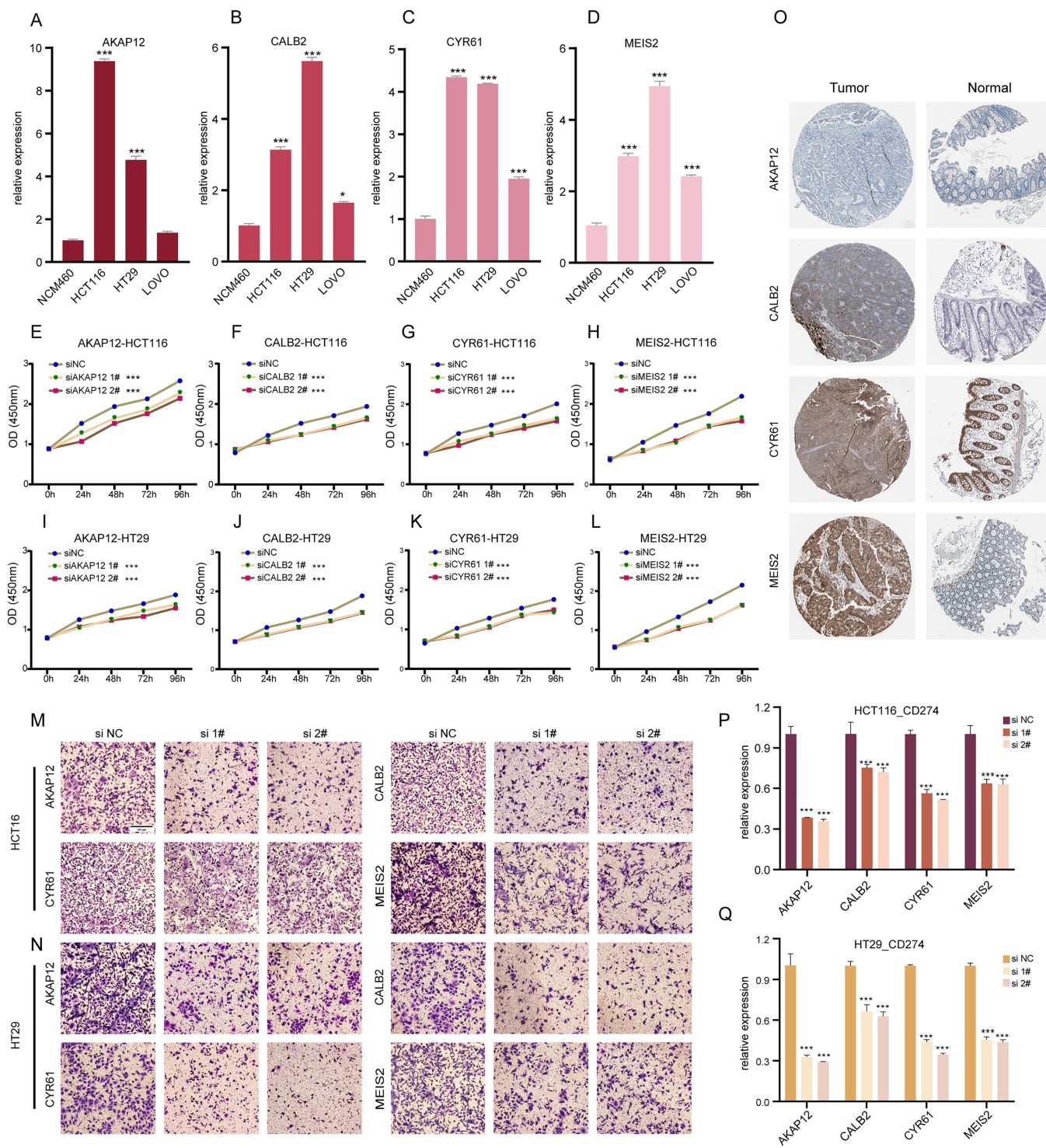

**Figure 6 Experimental validation of gene expression and functional assays.** (A–D) Assessing expression levels of AKAP12 (A), CALB2 (B), CYR61 (C), and MEIS2 (D) in normal colon cells (NCM460) and COADREAD cell lines (HCT116, HCT29, VOLO). (E–L) Post-knockdown viability reduction in HCT116 and HT29 cells were measured for AKAP12 (E, I), CALB2 (F, J), CYR61 (G, K), and MEIS2 (H, L). (M, N) Demonstrating decreased invasion capacity in HCT116 and HT29 cells following AKAP12, CALB2, CYR61, and MEIS2 knockdown. (O) Immunohistochemistry data from HPA database showed protein expressions of AKAP12, CALB2, CYR61, and MEIS2. (P, Q) Significant reduction in PD-L1 expression in HCT116 (P) and HT29 (Q) cells after knockdown of the respective genes. *p*-value denoted as ***$p < 0.001$.

## DISCUSSION

Colorectal adenocarcinoma, a leading malignancy globally, is witnessing a surge in incidence across various regions. Despite recent advancements, its heterogeneous nature and aggressive behavior pose challenges in prognostic assessment (*Miller et al., 2022*). Identifying novel biomarkers for personalized therapy development and prognosis improvement is thus a pressing need.

The pro-inflammatory immune microenvironment and neoantigen development in COADREAD have been linked to heightened immunotherapy response (*Zhang et al., 2022*; *Blass & Ott, 2021*). ICD involves pathways triggering intense inflammatory responses, potentially reshaping the tumor microenvironment and releasing tumor neoantigens (*Kroemer et al., 2022*). Given the high heterogeneity of colorectal cancers and specific molecular subtypes' sensitivity to immunotherapy (*Weng et al., 2022*), ICD offers insights into predicting and identifying patient groups suited for immunotherapy, including those with low response rates.

Therefore, exploring ICD in COADREAD presents a novel approach to understanding its microenvironmental impacts, prognostic significance, and predictive value. Addressing the unclear role of ICD in COADREAD research is vital for advancing therapeutic strategies and patient management.

As immunogenic cell death (ICD) emerges as a focal point in recent colorectal cancer research, a surge of studies has begun to further illuminate its pivotal role in enhancing patient-specific therapeutic strategies. *Ruan et al. (2020)* comprehensively reviewed the molecular events during the progression of colorectal cancer, focusing on their interactions and impacts on immunogenic cell death (ICD) under existing therapeutic interventions. This foundational work laid the groundwork for further exploration into the specific mechanisms by which ICD can influence treatment outcomes. Following this, *De Silva et al. (2024)* expanded on these concepts by systematically reviewing how inducers of ICD confer therapeutic benefits to colorectal cancer patients. They further discussed the potential of hallmark molecules of ICD to serve as biomarkers for both treatment and prognosis of colorectal cancer, underscoring their value in clinical applications. Building on these insights, *Lei et al. (2024)* focused on the practical applications of these molecular markers by developing and validating prognostic models that categorize colorectal cancer patients based on ICD. Our study extends this body of work by leveraging data from both single-cell and transcriptomic sequencing to develop a scoring system that categorizes colorectal cancer based on ICD. This scoring system not only aids in prognostic analysis but also enhances the evaluation of immunotherapy regimens, offering a new perspective on patient stratification and treatment optimization.

scRNA-seq stands out from traditional bulk RNA-seq by providing detailed insights into individual gene expression levels across cell subpopulations. This technique is instrumental in identifying specific biomarkers and understanding cellular heterogeneity in various cancers, including COADREAD (*Qi et al., 2022*).

In our study, we merged the insights gained from both bulk RNA-seq and scRNA-seq to perform a thorough analysis of COADREAD. This included examining ICD expression
profiles, conducting clustering analyses, assessing immune infiltration, exploring the mutation landscape, and screening for prognostic signature genes. The culmination of these analyses was the development of the ICD score, which has demonstrated considerable prognostic and predictive utility for immunotherapy responses in COADREAD.

In COADREAD, scRNA-seq analysis identified two ICD score clusters. The high-score cluster showed enriched inflammatory and TNFA signaling *via* NF-κB, indicating enhanced immune activation. Contrarily, these pathways were suppressed in the low-score cluster. Predominant in the high-score group were monocyte-macrophages and CD4+/CD8+ T cells, suggesting a robust anti-tumor immune response. These findings align with the concept that higher ICD scores correlate with more immunologically active "hot tumors."

We identified 12 ICD-related prognostic genes and divided COADREAD patients into two clusters based on these genes. Our analysis highlighted significant differences in prognosis, clinicopathological characteristics, immune infiltration, and pathway enrichment between the clusters, emphasizing ICD's role in regulating immune interactions in COADREAD.

In our study, we identified 448 differentially expressed genes (DEGs) between ICD score clusters in COADREAD, using univariate Cox regression analysis to pinpoint 25 OS-related genes. These genes, integral to the ICD score model developed *via* principal component analysis, included *SCG2*, *CALB2*, *HOXC6*, *CRYAB*, *ZNF532*, and others, each substantiating their roles in tumor progression, metastasis, and response to therapy.

For instance, *SCG2*, known for its prognostic value in immune infiltration and macrophage polarization, aligns with its role in chemotherapy and immunotherapy in colorectal cancer (*Weng et al., 2022*). *CALB2*'s modulation of 5-fluorouracil sensitivity (*Stevenson et al., 2011*), *HOXC6*'s association with poor OS and immunogenicity (*Qi et al., 2021*), and *CRYAB*'s involvement in CSC formation *via* the Wnt/β-catenin pathway (*Dai et al., 2022*) all highlighted their significance. Similarly, *ZNF532*'s role in oncogenic chromatin propagation in NUT midline carcinoma (*Alekseyenko et al., 2017*) and *AKAP12*'s influence on colon cancer metastasis (*Deng et al., 2022*) were noteworthy. Additionally, *FABP4*'s influence on colon cancer invasion (*Tian et al., 2020*), *MEIS2*'s role in cell migration (*Wan et al., 2019*), and *SLC2A3*'s correlation with prognosis and immune signature (*Gao et al., 2021*) were pivotal. Other genes like *SPARCL1*, *CYR61*, *CYP1B1*, and *COLEC12* have been implicated in various cancer dynamics, from tumor microenvironment modulation to disease progression and treatment response (*Shen et al., 2022*; *Xie et al., 2019*; *Chen et al., 2023*; *Tong et al., 2022*).

Our study, for the first time, reported the comprehensive effect of these ICD-regulated genes in COADREAD, offering new insights into their potential as therapeutic targets and prognostic markers in this complex disease.

Our study revealed stark differences in intrinsic properties between high and low ICD score groups in COADREAD. The high-score group, despite a poorer prognosis, exhibited more immune cell infiltration, a higher tumor mutation load, and elevated levels of immune checkpoint expression, including increased cytokines, chemokines, and their

receptors. Furthermore, a positive correlation was observed between the ICD score and immune-activated pathways.

Interestingly, the response to immunotherapy varied based on the ICD score. Patients in the high-ICD score cluster showed a potential benefit from ICIs and targeted therapies. This aligns with their higher immune mutational load and more active immune environment, as confirmed by previous studies highlighting the role of both immunotherapy and chemotherapy in inducing ICD and bolstering anti-tumor immunity (*Kroemer et al., 2022*; *Hayashi et al., 2020*; *Galluzzi et al., 2020*).

Our findings underscored that a high ICD score, indicative of increased immune cell infiltration and activated anti-tumor pathways, was predictive of a favorable response to immunotherapy. This suggested the close relationship between ICD status and the effectiveness of immunotherapy and other immunogenic treatments in COADREAD.

Our study faced limitations such as its retrospective nature and reliance on public sequencing data. Future studies should validate our findings in larger, prospective clinical trials, and include proteomic cross-validation for clinical applicability. Further basic research is needed to understand how ICD-related genes affect prognosis and response to therapy. Additionally, examining single-cell changes during anti-PD1 treatment could provide insights into treatment response heterogeneity in colorectal cancer (*Bassez et al., 2021*).

## CONCLUSIONS

In summary, our analysis of ICDRGs offers insights into the TIME and could guide future COADREAD research, particularly in drug development and personalized immunotherapy therapy.

## ACKNOWLEDGEMENTS

We are grateful to Yuanyuan Han, HuaShao for technical assistance and advice.

### Funding

The study was supported by the Medical Innovation Special Project of Shanghai Municipal Science and Technology Commission (22Y31920103) The funders had no role in study design, data collection and analysis, decision to publish, or preparation of the manuscript.

### Grant Disclosures

The following grant information was disclosed by the authors:
Medical Innovation Special Project of Shanghai Municipal Science and Technology Commission: 22Y31920103.

### Competing Interests

The authors declare that they have no competing interests.
## Author Contributions

- Jinglu Yu conceived and designed the experiments, performed the experiments, analyzed the data, prepared figures and/or tables, authored or reviewed drafts of the article, and approved the final draft.
- Yabin Gong analyzed the data, authored or reviewed drafts of the article, and approved the final draft.
- Zhenye Xu analyzed the data, authored or reviewed drafts of the article, and approved the final draft.
- Lei Chen performed the experiments, authored or reviewed drafts of the article, and approved the final draft.
- Shuang Li conceived and designed the experiments, analyzed the data, authored or reviewed drafts of the article, and approved the final draft.
- Yongkang Cui conceived and designed the experiments, analyzed the data, prepared figures and/or tables, authored or reviewed drafts of the article, and approved the final draft.

## Data Availability

The codes are available in the Supplemental Files.

## Supplemental Information

Supplemental information for this article can be found online at http://dx.doi.org/10.7717/peerj.17629#supplemental-information.

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
