# Peer review of "Prognostic and therapeutic insights into colorectal carcinoma through immunogenic cell death gene profiling"

_PeerJ, doi:10.7717/peerj.17629_

## Round 0.1 · original submission · Major Revisions

All three reviewers gave suggestions for revision. The authors are invited to do their best to revise the manuscript and answer questions.

**Language Note:** PeerJ staff have identified that the English language needs to be improved. When you prepare your next revision, please either (i) have a colleague who is proficient in English and familiar with the subject matter review your manuscript, or (ii) contact a professional editing service to review your manuscript. PeerJ can provide language editing services - you can contact us at copyediting@peerj.com for pricing (be sure to provide your manuscript number and title). – PeerJ Staff

Reviewer 1 ·

Basic reporting

This manuscript lays out a thorough investigation into the prognostic and therapeutic angles of Immunogenic Cell Death (ICD) in colorectal carcinoma with a keen eye on gene analysis and validation. The creation and validation of an ICD scoring system across different cohorts mark a significant leap forward. The study's approach, with its use of single-cell sequencing and tough statistical analyses, shows off its scientific muscle. Still, it could do with some buffing up in the stats department, especially around PCA and Cox regression, and making sure experimental checks are up to scratch. Also, a clearer connection between molecular subtypes and clinical outcomes, alongside a catchier title, would not go amiss. Some updates to the visuals could also make the data easier to get. I think it accepting this manuscript for publication after minor revisions.

Experimental design

1. The bit on siRNA transfection could do with some work to show how well the siRNA sequences did their job. Adding in some control experiments would make your results stand stronger.
2. For the cell culture part, spilling the beans on how many times the cell lines were passed around and keeping their characteristics consistent would help ease worries about variability...

Validity of the findings

1. In the analysis involving Principal Component Analysis (PCA) for developing the ICD score, we suggest diving deeper into the criteria for picking principal components. Including more on variance ratios or scree plot analysis might beef up the method's solidity.
2. The paper could really use a deeper dive into how the Cox regression model's assumptions were checked. It's pretty key to make sure the proportional hazards assumption is on point for the study's results to hold water.
3. Even though the paper does a solid job using plots to show off data, it's missing out on interactive or dynamic visualizations that could really boost understanding, especially for the tricky stuff like gene expression profiling. Throwing in some online extras with interactive features would be a big plus for the paper's impact.
4. The title you've got now covers the bases but could be polished up a bit to draw more attention and make things clearer. How about "Prognostic and Therapeutic Insights into Colorectal Carcinoma through Immunogenic Cell Death Gene Profiling" for a title that really hits the mark and highlights what's new?
5. Jumping from pinpointing molecular subtypes to proving the ICD score's chops for predicting how well immunotherapy works feels a bit jumpy. Laying out more clearly how these subtypes tie back to clinical outcomes and therapy responses would smooth things out.
6. When I was going through your methods and what you found on regulating the immune microenvironment, I stumbled on a study that seems super relevant. It's about how m6A regulator-mediated RNA methylation patterns play a part in the immune microenvironment in periodontitis. This could add some solid depth to your discussion on immunogenic cell death in colorectal carcinoma, given how RNA modifications mess with immune responses.

Reviewer 2 ·

Basic reporting

In this manuscript, Yu et al. explore Immunogenic Cell Death (ICD) in colorectal carcinoma. The study utilizes single-cell RNA sequencing to construct an ICD signature and analyzes TCGA and GEO data to identify two molecular subcategories. The aim is to offer potential anti-tumor strategies targeting ICD in colorectal carcinoma treatment. While the topic holds significant value, there are some concerns that should be properly addressed to improve the quality and robustness of the manuscript.

Experimental design

1. In the scRNA-seq dataset utilized by the study (GSE166555), other cell types, including mast cells, are present. Why they are not mentioned in the manuscript?

2. In the Materials & Methods section, Lines 93-96 indicate that the data was processed by harmony before the PCA. However, harmony (https://github.com/immunogenomics/harmony) suggests performing harmony on PCA embedding. Please explain why the data was processed in this way.

3. There were based on bulk RNAseq data. However, the manuscript fails to mention whether quality control of RNAseq/expression data was conducted.

Validity of the findings

1. In the Results section, the manuscript states the presence of a malignant cluster. Could the author(s) clarify how they identified malignant cells and distinguish them from normal cells?

2. The manuscript states at Line 266: 34 genes associated with ICD were identified using the "FindAllMarkers"? However, according to methods (Lines 107,108), the genes were selected from
a publication. Please clarify how the genes were identified precisely.

3. In the Results section, at Lines 309-312: please use statistical methods to determine if there are any significant differences between ICDRG expression and clinical features.

Additional comments

1. I recommend italicizing the gene names throughout the manuscript.
2. It is well know that cell cycle is associated with cell death (Clarke PR, Allan LA, Trends Cell Biol, 2009 Mar;19(3):89-98; Wiman KG, Zhivotovsky B, J Intern Med. 2017 May;281(5):483-495. ). I suggest author(s) investigate their relationships both in the scRNA and bulk-RNA data.

Reviewer 3 ·

Basic reporting

The language is intelligible and accurate in reporting and stating ideas with just a few errors. For instance, ‘a 5-year survival rate near 65%’ (Line 42) should be corrected as ‘a 5-year survival rate of near 65%’; the full form of the abbreviations should be spelt out when they first appear in the main text such as COADREAD (Line 46), ICDRGs (Line 75). Proofreading is advised.

With a funnel structure, the section of Introduction unfolds logically. The structure of the manuscript is clear. The methods, results and hypotheses correspond well to each other.

Experimental design

The manuscript focuses on the prediction of immunotherapy efficacy, which is a popular topic currently. Although lots of predicting models were established from different aspects, research gaps still exist. The research question of this manuscript is clearly defined and is meaningful.

The investigation was rigorously conducted by using bioinformatic methods and wet-lab experiments.

Methods were described with relatively adequate information, which helps other investigators to replicate.

Validity of the findings

Considering lots of predicting models were established from different aspects, the novelty of the manuscript is relatively limited, but it is still an addition to the field.

The data is robust and statistically sound, which leads to the conclusion in a reasonable way. The conclusion was stated properly.

Additional comments

I suggest the authors verify the model to pan-cancer to further test its robustness.

Another issue: COADREAD includes esophagus carcinoma? (Line 31) Please confirm!

Generally, in my opinion, I recommend this manuscript to be published in the prestigious journal after minor revision.

---

## Round 0.2 · Minor Revisions

There have been a number of recent studies of this kind using public databases for analysis, most of which have similar structures and do require authors to adequately cite and discuss the relevant literature.

The Section Editor noted:

> There are recent reviews of the data on ICD in colorectal carcinoma:

> Junping Lei, Jia Fu, Tianyang Wang, Yu Guo, Mingmin Gong, Tian Xia, Song Shang, Yan Xu, Ling Cheng & Binghu Lin (2024) Molecular subtype identification and prognosis stratification by a immunogenic cell death-related gene expression signature in colorectal cancer, Expert Review of Anticancer Therapy, DOI: 10.1080/14737140.2024.2320187

> De Silva, M., Tse, B.C.Y., Diakos, C.I. et al. Immunogenic cell death in colorectal cancer: a review of mechanisms and clinical utility. Cancer Immunol Immunother 73, 53 (2024). https://doi.org/10.1007/s00262-024-03641-5 I would ask the authors to address their data related to the review article and include this citation if appropriate.

> Also, consider these citations: Ruan H, Leibowitz BJ, Zhang L, Yu J. Immunogenic cell death in colon cancer prevention and therapy. Mol Carcinog. 2020 Jul;59(7):783-793. doi: 10.1002/mc.23183. Epub 2020 Mar 25. PMID: 32215970; PMCID: PMC7593824. Chiaravalli, M.; Spring, A.; Agostini, A.; Piro, G.; Carbone, C.; Tortora, G. Immunogenic Cell Death: An Emerging Target in Gastrointestinal Cancers. Cells 2022, 11, 3033. https://doi.org/10.3390/cells11193033 Yu, Chun, Yang, Weixuan, Tian, Li, Qin, Yue, Gong, Yaoyao and Cheng, Wenfang. "Construction of immunogenic cell death-related molecular subtypes and prognostic signature in colorectal cancer" Open Medicine, vol. 18, no. 1, 2023, pp. 20230836. https://doi.org/10.1515/med-2023-0836

> There appear to be many recent publications on this subject that are not cited, please review and explain how the present data is unique and informative.

Reviewer 1 ·

Basic reporting

After reviewing the revised manuscript, I'm convinced the changes have significantly improved its quality, meeting the publication standards. The effort to address previous concerns is commendable, and the manuscript now presents a solid case for acceptance.
I fully support its publication and look forward to its contribution to our field.

Experimental design

none

Validity of the findings

none

Reviewer 3 ·

Basic reporting

The authors have polished the language based on previous comments, corrected some minor grammatical errors and typos, and explained the abbreviations that appear for the first time. Overall, the language is fluent, clear, and easy to comprehend.

Experimental design

The authors have made some revisions to the methods section. Generally, the methods are described with relatively adequate information, which helps other investigators to replicate.

Validity of the findings

Similar to my previous opinion, the data is relatively robust and statistically valid, leading to the conclusion in a reasonable manner. The conclusion is stated appropriately.

Additional comments

I once suggested that the authors validate the model in pan-cancer. In fact, as the author replied, this would result in a significant increase in the complexity and volume of data. Furthermore, the article is mainly about colorectal cancer, so I understand the authors' decision not to conduct the validation in pan-cancer. I think it will not have a significant impact on the value of the manuscript.

Overall, I believe the article can be accepted and published.

---

## Round 0.3 · Minor Revisions

Although the reviewers all agreed to receive the manuscript, after reviewing it, I thought there were some things that could be improved.

1. "Figure" All photomicrographs should give a suitable field of view. (1) Micrographs of observation of cell population behavior should give a field of view that should be observed with more than 200 cells, such as the Transwell and scratch experiments. (2) The width of the micrograph of cell population behavior should not be less than 2.5 cm to ensure that the morphology of the cells can be clearly seen. (3) In the immunohistochemical staining of tissues, two fields of view should be provided, the large field should have more than 400 cells and the small field should have less than 100 cells, and the staining and localization can be clearly seen in small field. (4) All micrographs should be marked with scale.

Reviewer 2 ·

Basic reporting

After the author's revisions and editing, I am confident that the quality of the manuscript has been significantly improved to meet the publication standards. The authors have effectively addressed previous concerns, and the manuscript is now qualified for acceptance. I fully support its publication and look forward to its valuable contribution to our field.

Experimental design

The methods are sufficiently detailed, providing adequate information for replication.

Validity of the findings

The data demonstrates robustness and statistical validity, leading to well-stated conclusions.

Additional comments

None

---

## Round 0.4 · accepted · Accept

After revisions, all reviewers agreed to publish the manuscript. I also reviewed the manuscript and found no obvious risks to publication. Therefore, I also approved the publication of this manuscript.